# Should I study or should I go (to sleep)? The influence of test schedule on the sleep behavior of undergraduates and its association with performance

Ignacio Estevan[1]*, Romina Sardi[1], Ana Clara Tejera[1], Ana Silva[2], Bettina Tassino[3]

**1** Programa de Neuropsicología y Neurobiología, Facultad de Psicología, Universidad de la República, Montevideo, Uruguay, **2** Laboratorio de Neurociencias, Facultad de Ciencias, Universidad de la República, Montevideo, Uruguay, **3** Sección Etología, Facultad de Ciencias, Universidad de la República, Montevideo, Uruguay

* iestevan@psico.edu.uy

**Data Availability Statement:** Data are available from the project Open Science Framework database (https://osf.io/g7xfw/).

## Abstract

Sleep is crucial for college students' well-being. Although recommended sleep duration is between 7–9 hours per day, many students do not sleep that much. Scholar demands are among the causes of observed sleep deprivation in youth. We explored the influence of having a school test on previous night sleep in first-year students and the association of sleep duration and test performance. We ran two surveys in freshman students of the Universidad de la República, Montevideo, Uruguay: 1) 97 students of the School of Sciences who took the test at the same time; and 2) 252 School of Psychology students who took the test in four successive shifts. More than 1/2 of the participants (survey #1) and almost 1/3 (survey #2) reported short regular sleep duration (< 7h). In both samples, the sleep duration of the night before the test was reduced with respect to regular nights (survey #1: 2.1 ± 0.2 h, $p < 0.001$; survey #2: between 1.7 ± 0.4 h and 3.6 ± 0.3 h, all $p < 0.001$), with more than 10% of the students who did not sleep at all. In survey 2, sleep duration increased in later shifts ($F_{(3,248)} = 4.6$, $p = 0.004$). Using logit regressions, we confirmed that sleep duration was positively related to test scores in both samples (survey #1: exp B = 1.15, $p < 0.001$; pseudo-$R^2$ = 0.38; survey #2: exp B = 1.03, $p < 0.001$; pseudo-$R^2$ = 0.25). Delaying test start time may prevent the reduction in sleep duration, which may also improve school performance. In addition, educational policies should include information for students about the impact of sleep on learning and of the consequences of reduced sleep duration.

## Introduction

There is a consensus that adults should sleep between 7–9 hours per day [1, 2]. Chronic short sleep duration is associated with an increase in several risk factors [3], and with an increase in the relative risk for multiple-cause mortality [4]. However, according to the Centers for Disease Control and Prevention [5], more than 30% of American young adults report short sleep

**Funding:** IE was supported by a 2020-2023 Scholarship from Comisión Académica de Posgrado, Universidad de la República, Uruguay. The funders had no role in study design, data collection and analysis, decision to publish, or preparation of the manuscript.

**Competing interests:** The authors have declared that no competing interests exist.

duration (i.e. < 7h), and a recent meta-analysis showed a mean sleep duration among medical undergraduate students of 6.3 h per night [6].

It is well documented that biological and psycho-social changes during adolescence along with social pressures strongly influence high-school students' short sleep duration [7–9]. In particular, the compromise between the natural trend of adolescents towards Eveningness and their early school schedule has been pointed out as a key factor affecting sleep duration and quality [10, 11]. Young college undergraduate students still exhibit a delayed chronotype [12], and display a shorter sleep duration when attending morning classes with respect to evening classes [13].

Sleep is related to students' well-being and mental health [14–17]. Moreover, students' sleep duration, sleep pattern, and daytime sleepiness have been proved to affect their academic performance [18–21]. Comparing sleep problems to other factors influencing academic performance, Hartman & Prichard [22] found that sleep problems have similar influence in school retention and grades than other factors that receive more attention such as binge drinking or drug consumption.

Despite the relevance of sufficient sleep duration, there is some evidence that students reduce their sleep during exam periods and the night before a test [23]. An actigraphy study with final-year high school students showed a reduction in sleep duration, quality, and efficiency [24]. It has been interpreted that the increase in psychological distress and anxiety typical of exams periods affect both sleep duration and quality [23–27]. In addition to anxiety, using a survey with undergraduate students, Hartwig & Dunlosky [28] showed that more than half the participants often do all their study in one session previous to the test, and most of them also "cram" lots of information in this previous night sacrificing sleep hours with no benefits in their performance. In a study with American college students, almost 60% of them reported engaging at least once in all-night study sessions, which hampered their test performance [29]. Orzerch et al. [30] also found better grades among students not reporting all-night study sessions. However, other studies did not find an association between long night study sessions and test scores [31, 32].

Although the positive association of sleep duration and academic performance seems to be well established, the influence of tests on sleep is not yet fully understood, and studies about the consequences of an acute sleep reduction previous to a test are scarce. In this study, we took advantage of the highly populated freshman University courses of the Universidad de la República, Uruguay, that require students to take tests in different shifts. We aimed to clarify how students modify their sleep habits the night before they take a test and to evaluate the association between sleep duration and academic performance.

## Materials and methods

We ran two surveys in freshman students of the Universidad de la República, Montevideo, Uruguay. We used a short computer-based questionnaire to ask about students' sleep pattern (bedtime, sleep latency, sleep end) on regular nights and the night before a mid-term test that was applied immediately after students finished the test. Questionnaire also included some items about socio-demographic information.

Statistical analyses were performed in R [33] using RStudio as an integrated development environment [34]. Throughout the text sample statistics are reported as Mean ± Standard Deviation, while estimated differences are reported as Mean ± Standard Error. All procedures were approved by the Ethics Committee of the School of Psychology, Universidad de la República, and complied with the principles outlined by the Declaration of Helsinki [35]. All participants gave their informed consent to participate using a digital form.

## Survey #1

Undergraduate students of the first year semi-annual course of General Biology of the School of Sciences, Universidad de la República, were invited to participate in this study after finishing the mid-term test in May 2019. Ninety seven students agreed to participate (Table 1), representing the 33.5% of the students who took the test. The number of correct answers for each participant in the 20 multiple-choice questions test was provided by the course teachers. Test score represent 1/3 of the final grade.

## Survey #2

Undergraduate students of the Psychology School, Universidad de la República, who took the mid-term test of Neurobiology course in June 2019 were invited to participate in this study. Neurobiology is a first year semiannual course, and almost 2200 students (74.2% females) began their grade studies in Psychology in 2019 [36]. As 1358 students took the test, they were randomly distributed in four different shifts (see schedule in Table 1). Each one of the four test versions consisted of 60 similar true/false questions. Two hundred fifty two students agreed to participate (Table 1), representing the 18.6% of the students who took the test. In survey #2, a question inquiring about the total time spent studying previous to the test was added ("How many hours do you spend studying adding up yesterday and today hours previous to the test?"). The number of correct answers of each participant was provided by the course teachers. Test score represent 1/2 of the final grade.

## Results

Survey #1 sample was sex-biased towards females among participants (Table 1; $\chi^2$ = 8.7, $p$ = 0.003). Mean age was 21.0 ± 5.9. Regular reported sleep duration was 6.6 ± 1.5 h, and most participants reported regular sleep duration < 7h (56.7%). Sleep duration and pattern were modified the night before taking the General Biology test (Table 1, Fig 1A). Sleep duration was reduced in 2.1 ± 0.2 h (*paired-samples t-*test = 9.4, $p$ < 0.001), with 13 students (13.4%) being all-nighters. There was a moderate correlation between regular sleep duration and sleep duration before the test (r = 0.40, p < 0.001). Among sleepers, sleep onset was delayed in 1.0 ± 0.2 h (*paired-samples t-*test = 6.2, $p$ < 0.001), and sleep end was advanced in 0.5 ± 0.1 h (paired $t$ = 4.1, $p$ < 0.001). The delay in sleep onset was generated by an estimated delay of 1.1 ± 0.2 h in the bedtime (*paired-samples t-*test = -7.5, $p$ < 0.001), while sleep latency remained unchanged (*paired-samples t-*test = 0.0, $p$ = 1.0). A logit regression was used to study the

**Table 1. Characteristics of the participants of both studies.**

|  |  | Total | Males | Females | Age | Regular sleep duration | Sleep duration before test | All-nighters | Correct answer rate |
|---|---|---|---|---|---|---|---|---|---|
| Survey #1: School of Science | | | | | | | | | |
|  | 9:00 to 10:30 | 97 | 34 (35.1%) | 63 (64.9%) | 21.0 ± 5.9 | 6.6 ± 1.5 | 4.5 ± 2.4 | 13 (13.4%) | 0.67 ± 0.20[a] |
| Survey #2: School of Psychology | | | | | | | | | |
|  | 8:00 to 9:15 | 71 | 10 (14.1%) | 61 (85.9%) | 23.3 ± 7.3 | 7.7 ± 1.6 | 4.1 ± 2.4 | 12 (16.9%) | 0.60 ± 0.12[b] |
|  | 9:45 to 11:00 | 71 | 7 (9.9%) | 64 (90.1%) | 25.0 ± 7.9 | 7.9 ± 1.7 | 4.9 ± 2.5 | 9 (12.7%) | 0.64 ± 0.15[b] |
|  | 11:30 to 12:45 | 62 | 9 (14.5%) | 53 (85.5%) | 22.7 ± 6.8 | 7.4 ± 1.5 | 5.1 ± 2.8 | 11 (17.7%) | 0.65 ± 0.11[b] |
|  | 13:15 to 14:30 | 48 | 6 (12.5%) | 42 (87.5%) | 24.4 ± 8.8 | 7.7 ± 1.9 | 6.0 ± 2.5 | 5 (10.4%) | 0.68 ± 0.13[b] |

Note: In each case, time of test attendance is indicated. Discrete variables are presented as Number (Percentage); numeric variables are presented as Mean ± Standard Deviation.

[a] Test consisted of 20 multiple choice questions.

[b] Test consisted of 60 true/false questions.

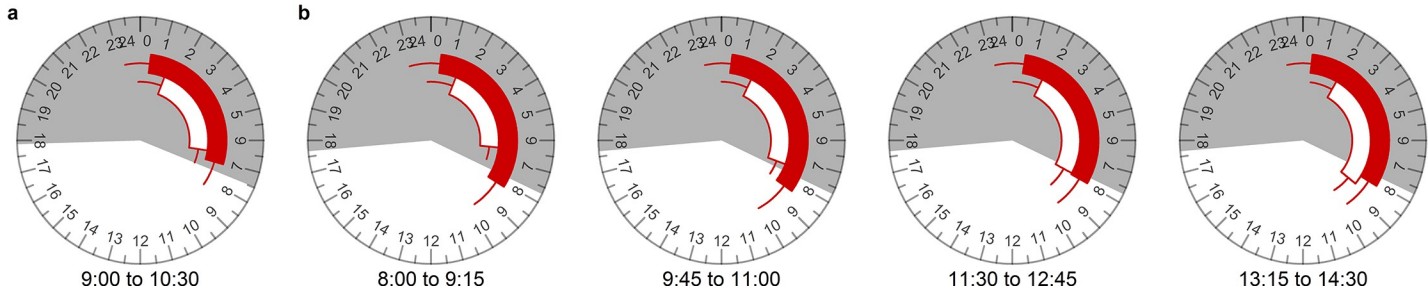

**Fig 1.** Sleep pattern among students that reported sleeping before the test from a) School of Science (survey #1); b) four test shifts in School of Psychology (survey #2).

association between the correct answer rate and sleep duration. Regular sleep was associated with the ratio of correct answers, and an hour increase in sleep duration was associated with a 10.8% increase in the odds ratio of correct answers ($z = 3.3$, $p < 0.001$; Cragg-Uhler pseudo-$R^2 = 0.11$). Sleep duration before the test was also a significant predictor, and an hour increase in sleep duration was associated with a 15.0% increase in the odds ratio of correct answers ($z = 6.8$, $p < 0.001$; Cragg-Uhler pseudo-$R^2 = 0.38$). The predicted correct answer rate was 52.3% for an all-nighter and 77.1% for a student who slept 8 h.

Sleep patterns are represented from Mean sleep onset to Mean sleep end (red lines represent the Standard Deviation) for regular-days sleep (red) and for the night before the test (white). Gray and white areas indicate photoperiod calculated from sunrise and sunset on the day before the test. Start time of the test is indicated below.

Survey #2 sample was also biased towards females (87.3%; $\chi^2 = 140.3$, $p < 0.001$), across shifts ($\chi^2 = 0.8$, $p = 0.8$). Mean age was 23.8 ± 7.7 years. Mean sleep duration on regular nights was 7.7 ± 1.6 h, with no differences between shifts ($F (3,248) = 1.1$, $p = 0.4$; Table 1). Among participants, 31.7% reported short regular sleep duration ($< 7$ h), with no difference between shifts ($\chi^2 = 5.3$, $p = 0.15$). Regular sleep pattern was also similar between shifts (mean sleep onset was 0:29 ± 1:23; mean sleep end was 8:10 ± 1:36), as no difference was found in either sleep onset ($F (3,248) = 0.2$, $p = 0.9$, Fig 1B) or sleep end ($F (3,248) = 0.7$, $p = 0.5$, Fig 1B). The night before the test sleep duration was reduced: sleep in 8:00 shift was reduced in 3.6 ± 0.3 h (*paired-samples t*-test $= 10.9$, $p < 0.001$), in 9:45 shift sleep was reduced in 3.0 ± 0.3 h (*paired-samples t*-test $= 9.1$, $p < 0.001$), in 11:30 shift was reduced in 2.3 ± 0.4 h (*paired-samples t*-test $= 6.6$, $p < 0.001$), and in 13:15 shift reduction was in 1.7 ± 0.4 h (*paired-samples t*-test $= 4.3$, $p < 0.001$). A small correlation between regular sleep duration and sleep duration before was observed (r = 0.09, p = 0.032). Sleep duration before the test was different between test shifts ($F (3,248) = 4.6$, $p = 0.004$; Table 1), as last shift students slept more than the first shift students (Tukey post-hoc $t = -3.8$, $p < 0.001$). All-nighters were 14.7% of participants, with no difference between test shifts ($\chi^2 = 1.7$, $p = 0.64$). Moreover, sleep reduction was different between test shifts ($F (3,248) = 4.9$, $p = 0.002$), and Tukey post-hoc test showed significant differences between students in 8:00 vs 11:30 shift ($t = 2.6$, $p = 0.048$), and between students in 8:00 vs 13:15 shift ($t = 3.6$, $p = 0.003$). Sleep pattern was also modified the night before the test (Fig 1B). Sleep onset the night before the test was similar between test shifts ($F (3,211) = 0.9$, $p = 0.7$), and was delayed compared to regular nights an estimated 1.3 ± 0.1 h (*paired-samples t*-test $= -12.6$, $p < 0.001$), with no difference between shifts ($F (3,211) = 0.5$, $p = 0.7$). This delay in sleep onset emerged from a delay in the bedtime of 1.3 ± 0.1 h (*paired-samples t*-test $= -11.9$, $p < 0.001$), similar between shifts ($F (3,211) = 0.6$, $p = 0.6$), as no difference was observed in the sleep latency compared to regular days (*paired-samples t*-test $= -0.5$, $p = 0.6$). Sleep end was dependent on the test start time ($F (3,211) = 39.0$, $p < 0.001$; Fig 1B): Sleep end was delayed as

test start later with all Tukey post-hoc paired comparisons significant (all $p < 0.035$). Sleep end difference between regular nights and the night before the test varied with test shift ($F(3,211) = 18.2$, $p < 0.001$): Sleep end was advanced $1.6 \pm 0.2$ h when test started at 8:00 (*paired-samples t*-test = 8.0, $p < 0.001$) and $1.1 \pm 0.2$ h when test started at 9:45 (*paired-samples t*-test = 5.3, $p < 0.001$), while no difference was observed in the other two shifts.

The association between the correct answer rate with the sleep duration and the test shift was studied using a logit regression model. Regular sleep duration did not predict test performance ($z = -1.45$, $p = 0.15$). However, an hour increase in sleep duration before the test was associated with a 3.8% increase in the odds ratio of correct answers ($z = 5.7$, $p < 0.001$; Cragg-Uhler pseudo-$R^2 = 0.12$). When the test shift was added it resulted in a significant predictor and model fit increased (Cragg-Uhler pseudo-$R^2 = 0.25$). Paired comparisons using Tukey adjustment showed a significant increase in the odd of correct answers in 9:45 shift (19.9%, $z = 4.1$, $p < 0.001$), 11:30 shift (22.3%, $z = 4.3$, $p = 0.001$) and 13:15 shift (36.3%, $z = 5.9$, $p < 0.001$) compared to 8:00 shift. The predicted correct answer rate was 56.7% for an all-nighter who attended the first shift and 69.0% for a student who slept 8 h and attended the fourth shift. Mean number of hours spent studying before the test was $8.0 \pm 5.0$ h, with no difference between test shifts ($F(3, 248) = 2.25$, $p = 0.08$). Number of hours spent studying did not correlate with sleep before the test ($r = -0.10$, $p = 0.3$) nor with test performance ($r = 0.03$, $p = 0.6$).

## Discussion

In this study, we present data to evaluate the influence of tests on freshman college students' sleep behavior and the influence of sleep on their academic performance. Although these issues have been addressed in previous reports [23, 24, 31, 32], this is the first study to explore how sleep patterns of the night before the test change when the test is taken at different times (survey #2). Overall (survey #1 and #2), most students delayed their bedtime the night before the test, reducing their sleep duration, and more than 10% did not even sleep at all the night before. Even when the test started as late as 13:15, the sleep duration of the night before was shorter than in regular nights, and 10% of the students stayed awake all night. In addition, sleep duration was positively correlated with the number of correct answers in the test, and therefore with school grades. When analyzing these effects across shifts (survey #2), we found that sleep duration and academic performance improved as test start times were delayed.

High rate of short regular sleep was found in both surveys, with more than 1/2 (survey #1) and almost 1/3 of students (survey #2) who reported sleep duration <7 h in average per night. The rate of students with short regular sleep was higher and mean sleep duration was shorter in students of School of Science (survey #1) than in students of the Psychology School (survey #2), probably because the former were younger than the latter [37]. Similar values of sleep deprivation were previously reported in Uruguayan university students [38], and this should be a matter of concern based on the multiple consequences of chronic inadequate sleep [14–17].

Taking the test had a strong influence in the night before sleep behavior of students, a pattern that has been previously described using both actigraphy data and questionnaires in young students [23, 24]. As stated by Hartwig & Dunlosky [28], sleep reduction seems to be a consequence of giving up hours of sleep to obtain more study hours prior to the test. We confirmed this general pattern in the present study as students of both surveys delayed their time to go to bed the night before the test in about 1 h, regardless of test start times.

Several previous studies reported a positive association between regular sleep duration and grades [32, 39–42], while others highlight the importance of regular sleep quality, rather than duration, on academic performance [39, 40, 43, 44]. We observed an association of regular sleep duration and test performance only in survey #1, as the odds ratio of correct answers

increased with sleep duration. Sleep duration on the night before the test did predict test performance in survey #2, and was a better predictor of performance in survey #1 compared to regular sleep. In Uruguay, grades use a non-linear scale from 0 to 12. The fair lowest passing grade is 3 and corresponds to 60% achievement, while 90% achievement corresponds to grade 10. This complexity of the Uruguayan grading system prevented us from using grades in regressions. However, in both surveys when predicted correct answer rate was converted to grades 8 h-sleepers obtained a passing grade while all-nighters did not. Scullin [45] found a similar result using actigraphy data and showing that long-sleep students outperform short-sleep students in tests scores. The difference between surveys in regression coefficients and explained deviance may be related with the different type of questions (true/false vs multiple choice) employed in both courses, as the probability of answering correctly at random is higher in true-false type questions.

School and test shifts, an obligated solution to the insufficient universities' infrastructure to deal with the progressively increasing number of students in many countries [46], can also be seen as an opportunistic tool to deepen the study of the influence of test start time on sleep and performance [47–49]. In survey #2, we found that the sleep duration of the night before the test increased as test start time was delayed. A previous study in Brazilian undergraduate students attending school in different shifts found a similar pattern [13]: sleep duration was longer in students of the afternoon-shift with respect to morning-shift ones. We also found that students' performance was significantly higher in later shifts with respect to early ones. The enhanced performance of late-shift students is more likely due to their longer sleep duration and not to the time spent studying the day before the test, which was not significantly different across shifts. In addition to longer sleep durations, chronotype-associated differences in performance may also contribute to the differences observed between shifts [50, 51]. To address this issue in the future, we plan to add the assessment of Morningness-Eveningness in students of the Psychology School taking tests in different shifts.

Our study has several limitations. Self-report questionnaires may overestimate sleep duration compared to objective measures [52], and short sleep prevalence may be even higher than reported among Uruguayan college students. Although data were collected immediately after the test to prevent memory blurring, future studies should include more objective measures to confirm our results. A previous study using actigraphy data found a similar pattern of sleep reduction during exam period [45]. Sleep disturbance and reduced performance may be both associated with the high levels of anxiety prior to a test [53]. However, we did not observe an increase in sleep latency before the test, a measure that has been related to anxiety levels [14, 54]. Nevertheless, the analysis of personality-linked variables could help to get a better understanding of the interaction between students' sleep behavior and study practice before a test.

In this study, we show that many college students reported not getting enough sleep. In addition, we found that taking a test influences students' sleep behavior, and that the sleep duration of the night before the test is associated with test performance. Given the relevance of adequate sleep, it appears as a cost-efficient way to improve student's academic performance and well-being [21, 55]. Although delaying school (and tests) start times has been related to longer sleep duration and better academic performance, it seems not enough. These evidence should inspire educational policies and promote an open communication of the impact of sleep on learning and of the consequences of reduced sleep duration.

## Acknowledgments

We thank everyone who participated in this study. We thank Álvaro Cabana for his suggestions for analysis.

## Author Contributions

**Conceptualization:** Ignacio Estevan.

**Formal analysis:** Ignacio Estevan.

**Investigation:** Ignacio Estevan, Romina Sardi, Ana Clara Tejera.

**Supervision:** Ana Silva, Bettina Tassino.

**Writing – original draft:** Ignacio Estevan.

**Writing – review & editing:** Ignacio Estevan, Ana Silva, Bettina Tassino.

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
