## [Decision Letter · Decision Letter 0]

27 Nov 2020

PONE-D-20-18113

Should I study or should I sleep?

PLOS ONE

Dear Dr. Estevan,

Thank you for submitting your manuscript to PLOS ONE. After careful consideration, we feel that it has merit but does not fully meet PLOS ONE’s publication criteria as it currently stands. Therefore, we invite you to submit a revised version of the manuscript that addresses the points raised during the review process. The reviewers has raised some concerns especially regarding the effect size and its significance which we wish to address in the revised version.

We look forward to receiving your revised manuscript.

Kind regards,

Mohammed Saqr, Ph.D

Academic Editor

PLOS ONE

Journal Requirements:

2. Please consider changing the title so as to meet our title format requirement (https://journals.plos.org/plosone/s/submission-guidelines). In particular, the title should be "Specific, descriptive, concise, and comprehensible to readers outside the field" and in this case it is not informative and specific about your study's scope and methodology.

3. Please clarify (on line 228) whether you can conclude about students' lack of sleep or their reported lack of sleep.

4. Please provide additional details regarding participant consent. In the ethics statement in the Methods and online submission information, please ensure that you have specified whether consent was informed.

Reviewers' comments:

Reviewer's Responses to Questions

**Comments to the Author**

1. Is the manuscript technically sound, and do the data support the conclusions?

Reviewer #1: Yes

Reviewer #2: Yes

2. Has the statistical analysis been performed appropriately and rigorously? 

Reviewer #1: Yes

Reviewer #2: I Don't Know

3. Have the authors made all data underlying the findings in their manuscript fully available?

Reviewer #1: Yes

Reviewer #2: Yes

4. Is the manuscript presented in an intelligible fashion and written in standard English?

Reviewer #1: Yes

Reviewer #2: Yes

5. Review Comments to the Author

Reviewer #1: This is an interesting study of university students and their sleep habits prior to a mid-term examination. the authors show that the sleep time the night before an exam is less than their habitual sleep and that grades correlated with the sleep time.

My main concern is that the size of the effect is small; in survey 1, 1 hour more sleep meant 5% increase in correct answers and in survey 1, 1 hour more sleep was only associated with 1.4% increase in correct answers. These numbers may not be significant to change the overall grade for the subject. Do the authors have data instead on whether more students failed if got less sleep? Or if the overall grade (ie, A to B, B to C) was effected because of less sleep?

Lines 185 and 186: not sure that 'short sleep ratio' is previously defined; would find different wording.

Reviewer #2: This is an important topic considered by the authors and studied in an ingenious manner i.e. in the group of students being offered exams in four different shifts. This allows the opportunity to test the effect of sleep duration and timing on outcomes of the test performance.

The manuscript is well written.

6. PLOS authors have the option to publish the peer review history of their article (what does this mean?). If published, this will include your full peer review and any attached files.

Reviewer #1: No

Reviewer #2: No

---

## [Author Response · Author response to Decision Letter 0]

22 Dec 2020

Academic Editor

The manuscript has been revised according to the journal style requirements.

Please consider changing the title so as to meet our title format requirement

We extended the title with a description of the aims of the study.

Please clarify (on line 228) whether you can conclude about students' lack of sleep or their reported lack of sleep.

Thank you for the observation. We modified the expression to better describe our findings (line 241).

Please provide additional details regarding participant consent. In the ethics statement in the Methods and online submission information, please ensure that you have specified whether consent was informed.

The use of informed consent was included in line 90.

We note that you have stated that you will provide repository information for your data at acceptance. Should your manuscript be accepted for publication, we will hold it until you provide the relevant accession numbers or DOIs necessary to access your data.

The information was included in the Cover Letter.

Reviewer #1

My main concern is that the size of the effect is small; in survey 1, 1 hour more sleep meant 5% increase in correct answers and in survey 1, 1 hour more sleep was only associated with 1.4% increase in correct answers. These numbers may not be significant to change the overall grade for the subject. Do the authors have data instead on whether more students failed if got less sleep? Or if the overall grade (ie, A to B, B to C) was effected because of less sleep?

Thank you for this observation. We did again the analyses using logistic regressions to explain the correct answers rate. The results did not change, but these new analyses explain a greater proportion of the deviance, better modeling the association between sleep and performance (lines 124-131 and 168-177 in Results, table 1, and lines 208-211 in Discussion). The abstract was modified accordingly (lines 32-34).

In Uruguay, grades are given in figures on a scale from 0 to 12.The fair lowest passing grade is 3 and corresponds to 60% achievement, while 91% achievement corresponds to grade 10. This complexity of the Uruguayan grading system prevented us from including the grades. Instead, we estimated the correct answer rate for extreme sleep durations of 0h and 8h (lines 130-131 and 176-177). The smaller change in survey #2 is now discussed in lines 213-216.

Lines 185 and 186: not sure that 'short sleep ratio' is previously defined; would find different wording.

The expression was replaced (lines 194-195).

---

## [Decision Letter · Decision Letter 1]

4 Jan 2021

PONE-D-20-18113R1

Should I study or should I go (to sleep)? The influence of test schedule on the sleep behavior of undergraduates and its association with performance.

PLOS ONE

Dear Dr. Estevan,

Thank you for submitting your manuscript to PLOS ONE. After careful consideration, we feel that it has merit but does not fully meet PLOS ONE’s publication criteria as it currently stands. Therefore, we invite you to submit a revised version of the manuscript that addresses the points raised during the review process. Please amend the discussion with needed explanations.

We look forward to receiving your revised manuscript.

Kind regards,

Mohammed Saqr, Ph.D

Academic Editor

PLOS ONE

Reviewers' comments:

Reviewer's Responses to Questions

**Comments to the Author**

1. If the authors have adequately addressed your comments raised in a previous round of review and you feel that this manuscript is now acceptable for publication, you may indicate that here to bypass the “Comments to the Author” section, enter your conflict of interest statement in the “Confidential to Editor” section, and submit your "Accept" recommendation.

Reviewer #1: All comments have been addressed

2. Is the manuscript technically sound, and do the data support the conclusions?

Reviewer #1: Yes

3. Has the statistical analysis been performed appropriately and rigorously? 

Reviewer #1: Yes

4. Have the authors made all data underlying the findings in their manuscript fully available?

Reviewer #1: Yes

5. Is the manuscript presented in an intelligible fashion and written in standard English?

Reviewer #1: Yes

6. Review Comments to the Author

Reviewer #1: The authors have better explained the grading system in Uruguay and why they cannot provide 'grades' per se. This explanation would also be helpful to other readers and therefore my final recommendation is that the sentences written to me about grading be added to the Discussion. I think important as clearly the predicted grades on line 131 represent 'failure' v. 'passing.'

7. PLOS authors have the option to publish the peer review history of their article (what does this mean?). If published, this will include your full peer review and any attached files.

Reviewer #1: No

---

## [Author Response · Author response to Decision Letter 1]

29 Jan 2021

As suggested, sentences describing the grading system in Uruguay were included in the discussion.

---

## [Editor Report · Decision Letter 2]

2 Feb 2021

Should I study or should I go (to sleep)? The influence of test schedule on the sleep behavior of undergraduates and its association with performance.

PONE-D-20-18113R2

Dear Dr. Estevan,

We’re pleased to inform you that your manuscript has been judged scientifically suitable for publication and will be formally accepted for publication once it meets all outstanding technical requirements.

Kind regards,

Mohammed Saqr, Ph.D

Academic Editor

PLOS ONE

---

## [Editor Report · Acceptance letter]

15 Feb 2021

PONE-D-20-18113R2 

Should I study or should I go (to sleep)? The influence of test schedule on the sleep behavior of undergraduates and its association with performance. 

Dear Dr. Estevan:

I'm pleased to inform you that your manuscript has been deemed suitable for publication in PLOS ONE. Congratulations! Your manuscript is now with our production department. 

Kind regards, 

on behalf of

Dr. Mohammed Saqr 

Academic Editor

PLOS ONE